

**Influence of the melting temperature on the measurement**
**of the mass concentration and size distribution of black**
**carbon in snow**
**T. Kinase[1], K. Kita[1], Y. Ogawa[2], and K. Goto-Azuma[2,3]**
[1]{Graduate school of Science and Technology, Ibaraki University, Ibaraki, Japan}
[2]{National Institute of Polar Research, Tokyo, Japan}
[3]{SOKENDAI(The Graduate University of Advanced Studies)}
Correspondence to: T. Kinase (10nd402l@vc.ibaraki.ac.jp)
K. Kita (kazuyuki.kita.iu@vc.ibaraki.ac.jp)
Y. Ogawa (ogawa.yoshimi@nipr.ac.jp)
K. Goto-Azuma (kumiko@nipr.ac.jp)

## 14 Abstract

The influence of temperature and time for the melting snow samples on the measurement of
mass concentration and its size distribution of black carbon (BC) in snow was evaluated with
experiments. In the experiments, fresh (Shirouma) and aged (Hakusan) snow samples were
melted at different temperatures or time conditions, and BC mass concentration and its size
distribution in the melted snow samples were measured with a nebulizer and a single particle
soot photometer (SP2). The experiment changing the melting temperature conditions
indicated that BC mass concentration in the liquid decreased at high melting temperature of
70 ℃. The decrease was 9.1 % for the Shirouma sample and 42.3 % for the Hakusan sample.
This decrease depended on the BC particle size: significant decrease was found at BC
diameter less than 350 nm. Similar decrease of the BC mass concentration was also found
when the Hakusan snow sample melted at 5 ℃ was heated to 70 ℃. The experiment changing
the melting time condition indicated that BC mass concentration in the liquid did not change
for the Shirouma sample, and that it decreased significantly with longer melting time for the
Hakusan sample (37.7 %). These results indicated that the snow sample melting at a high



temperature or in a long time can significantly affect the measurement of BC mass and its size
distribution, especially for aged snow samples.
**1    Introduction**
Black carbon (BC), which is commonly referred as soot, strongly absorbs the solar radiation
in the atmosphere, leading to large climate effects. BC deposited on the snow/ice packs also
leads to positive radiative forcing, because it can reduce the snow/ice albedo significantly
(Warren and Wiscombe, 1980; Wiscombe and Warren, 1980; Bond et al., 2012). The
snow/ice albedo effect of BC also accelerates the melting of snow and ice (Aoki et al, 2011;
Brandt et al, 2011). Therefore, many studies have been carried out to measure BC
concentration in snow/ice and have estimated its influence on the climate via the radiative
forcing, the surface albedo and so on.
In the latest report by the Intergovernmental Panel on Climate Change (IPCC, 2013), the
global radiative forcing of BC in snow since 1875 was estimated to be 0.04 $Wm^{-2}$ with a 90%
uncertainty range between +0.02 and +0.09 $Wm^{-2}$. Bond et al. (2012) also estimated it to be
between +0.01 and +0.09 $Wm^{-2}$. These works showed that there remains a large uncertainty in
estimating climate effects of BC on snow/ice, more studies are necessary for BC in snow,
including the improvement of its measurement techniques.
BC mass concentration in snow/ice is commonly measured with the following three methods.
First one utilizes the light absorption by BC. In this method, snow sample is melted and
filtered. Diffuse light is irradiated to the filter to measure its light transparency, and the mass
of BC retained on it is derived from decrement of the transparency (Clarke and Noone, 1985;
Warren and Clarke, 1990; Grenfell et al, 2011). Second one utilizes the thermal optical
technique. In this method, snow sample is also melted and filtered with a quartz fiber filter.
BC retained on it are thermally converted to $CO_2$ gas, and the mass of BC is derived by
measuring the $CO_2$ concentration (Chow et al, 2001; Aoki et al, 2011; Chow et al., 2007).
Third one utilizes the single particle soot photometer (SP2). In this method, snow samples are
melted and aerosolized by a nebulizer. Mass and size of each aerosolized BC particles are
measured with the SP2 instrument (McConnell et al, 2007; Schwarz et al, 2012; Schwarz et al,
2013; Ohata et al, 2013; Lim et al, 2014). This method has an advantage: it can provide not
only the total mass of BC but also the size distribution of BC particles in snow. The BC size
distribution is very important because it affects the mass absorption cross section (MAC) of





BC in snow significantly (Schwarz et al, 2013). There are several factors affecting
uncertainties of the measured BC mass concentration in these methods. For example, Schwarz
et al. (2012) and Ohata et al. (2013) estimated uncertainties in the SP2 method by evaluating
the size-dependent efficiency of a nebulizer and the effect of agitation of BC samples in water
solution.
Although the melting of snow/ice samples is a common procedure in the above three methods,
the uncertainties due to this procedure has not fully examined so far. Many authors use a
microwave oven or a hot water bath to melt the snow/ice samples faster to reduce BC loss at
inner wall surface of sample containers and to reduce variation of BC size distribution in them
(e.g. Warren and Clarke, 1990; Doherty et al, 2010; Schwarz et al, 2012; Brandt et al, 2011).
However, some studies suggest that the heating condition for melting snow/ice sample may
affect the BC mass concentration and its size distribution in the melted water significantly,
leading possible uncertainty in the measurement of BC mass in snow/ice. Schwarz et al.
(2012) and Lim et al. (2014) indicate that freezing and melting cycles affect BC mass
concentration and its size distribution in water significantly, and that it is possible that BC
size distribution can give information about the thermal history of the snow. Lim et al. (2014)
evaluated for sample treatment procedures in the SP2 method, including melting procedure,
freezing/melting cycle, surface area to volume ratio of sample containers. They compared two
melting procedures, melting snow samples at room temperature and in a warm bath (30 ℃),
and did not find a significant difference in the measure BC concentration. They suggested that
the faster melting of snow samples would be better, and that the melting temperature would
not affect the measurement significantly. However, considering that higher temperature has
been adopted for the faster melting in snow/ice samples with a microwave oven in several
studies, the temperature influence should be evaluated for wider temperature range. It is
possible that history of snow/ice samples, such as various chemical species deposited on snow
or cycles of partially melting and refreezing, affect the influence of the melting conditions on
the measurement. In this study, by using fresh and aged snow samples, experiments to
measure BC mass and its size distribution in the samples melted with various temperature and
time conditions were conducted with the SP2 instrument (Droplet Measurement Technology,
Boulder, Colorado, USA). Comparing results for the different temperature or melting time
conditions, their influences on the BC measurement in snow/ice are evaluated.





## 2   Experiment
### 2.1   Snow samples
The snow samples used in this study were obtained at two locations, Hakusan and Shirouma,
Japan, as shown in Fig.1. The sampling date and latitude/longitude of the two locations are
listed in Table.1. The Shirouma sample was fresh, powder snow, sampled within 6 hours from
the snowfall. The Hakusan sample consisted of aged, granular snow which was sampled about
2 weeks after the snowfall. This sample experienced partial melting by sunlight heating or
rainfall and refreezing. After the sampling, these samples had been kept in polypropylene
(PP) containers at temperatures below -30 °C. In order to make each snow sample more
uniform, it was well stirred with a mixer in the low temperature laboratory (room temperature
was -30 °C) in National Institute of Polar Research before its division and melting.
We measured some chemical compositions with ion chromatographs (IC-2010, Tosoh Co.,
Ltd., Tokyo, Japan) for both samples. Concnetrations of $SO_4^{2-}$, $NO_3^-$, $Na^+$ and $Cl^-$ were
significantly different between Hakusan and Shirouma samples: their concentrations were
respectively 0.85, 0.59, 0.74 and 0.80 ppm in the Hakusan sample, and 0.15, 0.12, 0.47 and
0.29 ppm in the Shirouma sample. Larger concentrations of $SO_4^{2-}$ and $NO_3^-$ indicate that the
Hakusan snow sample contained more pollutants in comparison with the Shirouma snow
sample.
### 2.2   Melting process
In this study, two experiments were performed to evaluate the influences of the melting
temperature and the melting time. For the experiment to evaluate the melting temperature
influence (melting temperature experiment), each (Hakusan or Shirouma) snow sample was
stirred in a 500 cm$^3$ bottle and was divided into nine 30 cm$^3$ grass bottles before melting.
Snow samples in each of the three bottles were melted in a water bath at three temperatures of
5 °C, 20 °C and 70 °C. Inhomogeneity in each snow sample was estimated with the standard
deviation of measurement results for these three bottled samples melted at a same temperature.
For the experiment to evaluate the melting time influence (melting time experiment), a larger
amount of snow sampled in a wider area at Shirouma or in a larger depth range at Hakusan
was used in comparison with the melting time experiment. Therefore, BC mass concentration



in each snow sample was not necessary identical in these two experiments. In the melting
time experiment, each snow sample was stirred in a 6000 cm$^3$ metallic can and was divided
into three 30 cm$^3$ grass bottles and into one 500 cm$^3$ grass bottle. Snow samples in these
bottles were melted in a refrigerator at a temperature of approximately 1 °C. The difference in
the bottles volume changed the melting time; it took 2–3 hours and more than 6 hours for the
melting of the snow samples in the 30 cm$^3$ and 500 cm$^3$ bottles, respectively. A low
temperature of 1 °C was adopted because we had found that a lower melting temperature is
better for the snow BC measurement in the melting temperature experiment, and because a
lower melting temperature could increase the time difference in the melting time of the snow
sample.
**2.3   Measurement of the BC mass concentration**
The melted snow samples were sonicated for 15 minutes, and the mass concentration of BC
and its size distribution in them were measured with the experiment system similar to that
used in Ohata et al. (2011 and 2013). The liquid of the melted snow sample was transferred to
a concentric pneumatic nebulizer (Marin-5, Cetac Technologies Inc., Omaha, Nebraska, USA),
with a peristatic pump (REGRO Analog, ISMATEC SA., Feldeggstrasse, Glattbrugg,
Switzerland) to be aerosolized with dry, filtered air flow at a flow rate of 15.23 cm$^3$ s$^{-1}$. The
droplets, converted from the sample liquid, in the air flow from the nebulizer were evaporated
while passing through a heater tube at 140 ℃ and the water vapor in the air flow was removed
while passing through a chiller at 3 ℃. BC and other non-volatile particles remained in the air
flow were introduced to the SP2 to measure mass and size of each BC particles. The details of
the used SP2 instrument have been described elsewhere (Stephen et al, 2003; Moteki and
Kondo, 2007; Moteki and Kondo, 2010).

**3   Results**
**3.1   Influence of the melting temperature**
Figures 2(a) and 2(b) show the total mass concentration of BC in the melted Hakusan and
Shirouma snow samples, respectively. The total mass concentrations of BC in the Hakusan
snow sample melted at temperatures of 5 ℃, 20 ℃ and 70 ℃ are 61.9±7.4, 59.7±10.1 and
35.7±6.2 µg L$^{-1}$, respectively, and they are 25.7±3.1, 25.4±1.7 and 23.4±0.6 µg L$^{-1}$ in the



Shirouma snow sample, respectively. For both samples, the BC mass concentration values do
not show a significant difference between the melting temperatures of 5 ℃ and 20 ℃. On the
contrary, those values are systematically smaller at the melting temperaturesof 70 ℃. In the
case of the Hakusan sample, the difference between 70 ℃ and 5/20 ℃ is significant,
exceeding random errors in the measurement and sample inhomogeneity.
Figures 3(a) shows mass size distribution of BC in the Hakusan snow samples melted at
temperatures of 5 ℃, 20 ℃ and 70 ℃, and Figure 3(b) is the same but for the Shirouma snow
sample. For the Hakusan sample, the mass concentrations of the BC particles of diameters
less than 400 nm is significantly smaller in the snow melted at 70 °C than those at the other
temperatures. For the Shirouma sample, the BC mass size distribution at the three melting
temperature are similar with each other, but the values at higher melting temperature are
smaller for the BC particles of diameter less than 350 nm. These results show that the BC
mass concentration in the melted snow decreases at higher melting temperatures as 70 ℃,
especially for BC particles smaller than 350 nm. Figure 4 shows the size distributions of the
BC mass ratio of the 70 °C melting sample to the 5 °C melting sample, indicating that the
ratio systematically decreases with the decrease of the BC particle diameter. The difference
between the Shirouma and Hakusan samples suggests that this decrease presumably depends
on the snow age and/or amount of impurities in snow, considering that the Hakusan sample
was more aged and that it contained more pollutants such as $SO_4^{2-}$ and $NO_3^-$ in comparison
with the Shirouma sample.
The influence of high temperature (or heating) is also evaluated by another experiment. We
heated the liquid of the Shirouma snow sample melted at 5 °C to the temperature at 70 °C,
and measured the mass size distribution of BC. Figure 5 shows the size distribution of the BC
mass ratio between the before and after the heating. The error bars show reproducibility of the
measurement. The ratio decreased significantly with BC diameter smaller than 300 nm,
indicating that reduction of the mass concentration for smaller BC particles can occur at
higher temperature around 70 °C not only during the snow melting but also after the melting.
**3.2   Influence of the melting time**
Figure 6(a) and 6(b) show the total mass concentration of BC in the melted Hakusan and
Shirouma snow samples, respectively, comparing those values in the sample melted in the 30
$cm^3$ bottles and in the 500 $cm^3$ bottles. The time required for the melting was about 2 hours in





the 30 cm$^3$ bottles, and it exceeded 6 hours in the 500 cm$^3$ bottles. The total mass
concentration of BC in the Hakusan snow sample melted in the 30 cm$^3$ bottles and in the 500
cm$^3$ bottles are 36.8±2.7 µg L$^{-1}$ and 22.9±2.3 µg L$^{-1}$, respectively, the difference is significant,
exceeding randam error range. On the contrary, in the Shirouma snow sample, they are
16.3±1.9 µg L$^{-1}$ and 15.1±2.5 µg L$^{-1}$, respectively, and the difference between the two is
negligibly small. In this experiment, the sum of the random error in the measurement and the
inhomogeneity of snow is estimated with the standard deviation of measured values on three
bottle samples for the 30 cm$^3$ bottles.  For the 500 cm$^3$ bottle sample, because only a single
bottle sample was measured, the random error range was estimated as a root mean sum of the
standard deviation of measured values of the single 500 cm$^3$ bottle and standard deviation of
three bottle samples for the 30 cm$^3$ bottles to include an error due to the sample
inhomogeneity.
Figure 7(a) shows mass size distribution of BC in the Hakusan snow samples melted in the 30
cm$^3$ bottles and in the 500 cm$^3$ bottle, and Figure 7(b) is the same but for the Shirouma snow
samples. The BC mass concentration of smaller BC particles in the 500 cm$^3$ bottle sample are
less than those in the 30 cm$^3$ bottle sample both for the Hakusan and Shirouma samples.
However, the difference in the Shirouma samples was not significant considering the error
range. On the other hand, the difference in the Hakusan samples was significantly smaller at
diameters less than 850 nm in the 500 cm$^3$ bottles. Figure 8 shows the size distributions of the
BC mass ratio of the 500 cm$^3$ bottle sample to 30 cm$^3$ bottle samples. The ratio is
systematically smaller than 1 at the BC diameters between 100 nm and 350 nm for the
Shirouma sample, but it was not significant. The ratio is significantly smaller than 1 at the BC
diameters less than 850 nm for the Hakusan sample but there is no size dependence. These
results suggest that it is possible that reduction of the mass concentration for BC particles at
70 to 850nm can occur during the melting process especially for aged and/or more polluted
snow. Dependence on the BC diameter of the reduction was smaller than that in the melting
temperature experiment.
It is possible that this time-dependent decrease of the BC mass during the snow melting
continues after it. We kept the melted Hakusan and Shirouma samples in the 500 cm$^3$ bottles
at 1 °C in the refrigerator for 4 days since the melting, and measured the BC mass repeatedly
with a time interval of about 24 hours to evaluate the influence of the storage time length. No
significant difference was found during time progress both in the total mass and the size





distribution, indicating that the significant BC mass reduction does not occur in the liquid
phase during storage at the low temperature after the melting. This result is consistent with the
recent reports; Schwartz et al. (2013) and Lim et al. (2014) showed that the size distribution
of BC mass in melted snow samples is stable over time range of 24 hours, and Ohata et al.
(2011) showed that the BC mass concentration in water solution did not significantly change
over 50 days. It is suggested that the BC mass reduction in the melting experiment occurred
during coexistence of solid and liquid phases.
**4   Conclusions**
In this study, for more accurate measurement of BC amount in snow, we evaluated the
influences of the temperature and the time in the procedure of melting snow samples.
Changing the temperature (melting temperature experiment) or the time (melting time
experiment), we melted snow samples to measure BC mass concentration and its size
distribution in them with the SP2 method. Fresh (Shirouma) and aged (Hakusan) snow
samples were used in these experiments, and each snow sample was stirred well before the
experiments to reduce its inhomogeneity.
In the melting temperature experiment, each snow sample was divided and melted at
temperatures of 5 ℃, 20 ℃ and 70 ℃ in a water bass. For both Hakusan and Shirouma snow
samples, the measured values of total BC mass concentration in liquid melted at the
temperatures of 70 ℃ were smaller than those at 20 ℃ and 5 ℃. The difference between 70
℃ and 5 ℃ in the Hakusan sample was larger (42.3 %) than that (9.1 %) in the Shirouma
sample, and it exceeded the random error range. This systematic decrease of the mass
concentrations at the melting temperature of 70 ℃ occurred in BC particles of diameters less
than 350 nm. Similar decrease of the BC mass concentrations was also found when the snow
sample liquid melted at 5 ℃ was heated to 70 ℃. These results indicate that the decrease by
the heating to high temperature can occur not only during the snow melting but also during
the storage in the liquid phase.
In the melting time experiment, the Hakusan and Shirouma snow samples in the 30 cm$^3$
bottles were melted for about 2 hours, and those in the 500 cm$^3$ bottles were melted for more
than 6 hours. In the case of the Shirouma snow sample, measured values of the BC mass
concentration in liquid melted in 30 cm$^3$ and 500 cm$^3$ bottles were nearly the same. In contrast,
that value in liquid melted in 500 cm$^3$ bottle was significantly (about 37.7%) smaller than that




in 30 cm$^3$ bottles in the case of the Hakusan snow sample. Their decreasing rate in the 500
cm$^3$ bottle was nearly a constant at BC particle diameters less than 850 nm. Significant BC
mass reduction does not found in the liquid phase during storage at the low temperature after
the melting, suggesting that the decrease occurred during coexistence of solid and liquid
phases.
The experimental results in this study show that snow samples should be melted at a lower
temperature during a short time for reduction of uncertainty in the measurement of BC mass
in snow. The melting temperature experiment showed that it is possible that the heating to a
high temperature cause a significant reduction of BC both during the melting and in the liquid
phase. The melting time experiment showed that slow reduction of BC can occur even at a
low temperature under the coexisting of solid and liquid phases. Because significant decrease
of BC in the melted snow occurred in the Hakusan snow samples both in the melting
temperature and melting time experiments, the influences of the melting temperature and the
melting time on BC mass would be more significant in aged and/or polluted snow.
In this study, mechanisms of the BC decrease during the snow melting were not examined.
The BC mass decrease can be caused by adsorption of BC at inner wall of bottles in which the
snow samples were melting.  It is also possible that agglomeration of smaller BC particles to
super-micron BC, which was not measured here, causes the apparent decrease of BC mass,
considering that the decrease of BC mass is more significant at smaller BC diameters in the
melting temperature experiment. It should be noted that the size-dependence of BC decrease
was different in the melting time and melting temperature experiments, suggesting that
multiple mechanisms contributes to the BC decrease during the snow melting. Because the
BC reduction at 5 ℃ was found only during the snow melting, it may occur at interface
between solid and liquid phases. We suspect that contaminating chemical substances may
play a part in this BC decrease. Schwarz et al. (2012) showed that the mixing of nitric acid to
liquid samples causes reduction in the mass fraction of super–micron BC particles. On the
contrary, Ohata et al. (2011) showed that there is no significant effect for $(NH_4)_2SO_4$ and
Suwannee River Fulvic Acid (SRFA) at their typical amounts in Tokyo, except for much
higher concentration. However, the influence of impurities has not fully understood, and
influences of the organic/inorganic substances in snow should be studied more detailed.




**Acknowledgements**
This work is supported by National Institute of Polar Research and by Ibaraki University. The
authors would like to thank S. Ohata, N. Moteki, H. Motoyama, M. Shiobara, Y. Zaizen, and
K. Adachi for the experimental supports, and Y. Kondo, M. Hayashi, K. Hara and K. Kuchiki
for the discussions.

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

Table 1. Information on snow samples.

| Sampling Location | Latitude | Longitude | Sampling Date | age of snow |
|---|---|---|---|---|
| Hakusan | 36.174 | 136.627 | 2013.3.13 | about 2 weeks(coarse-grained old snow) |
| Shirouma | 36.764 | 137.882 | 2013.3.21 | less than 6 hours (powder snow) |



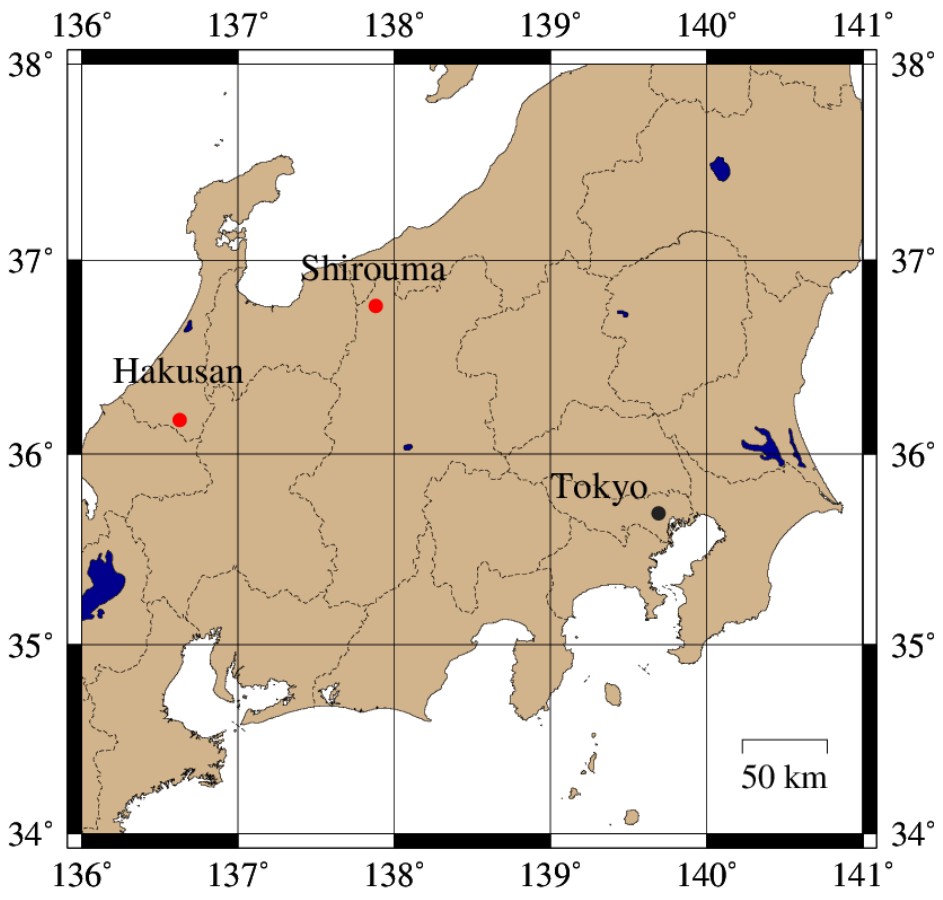

2    Figure 1. A map showing snow sampling locations.





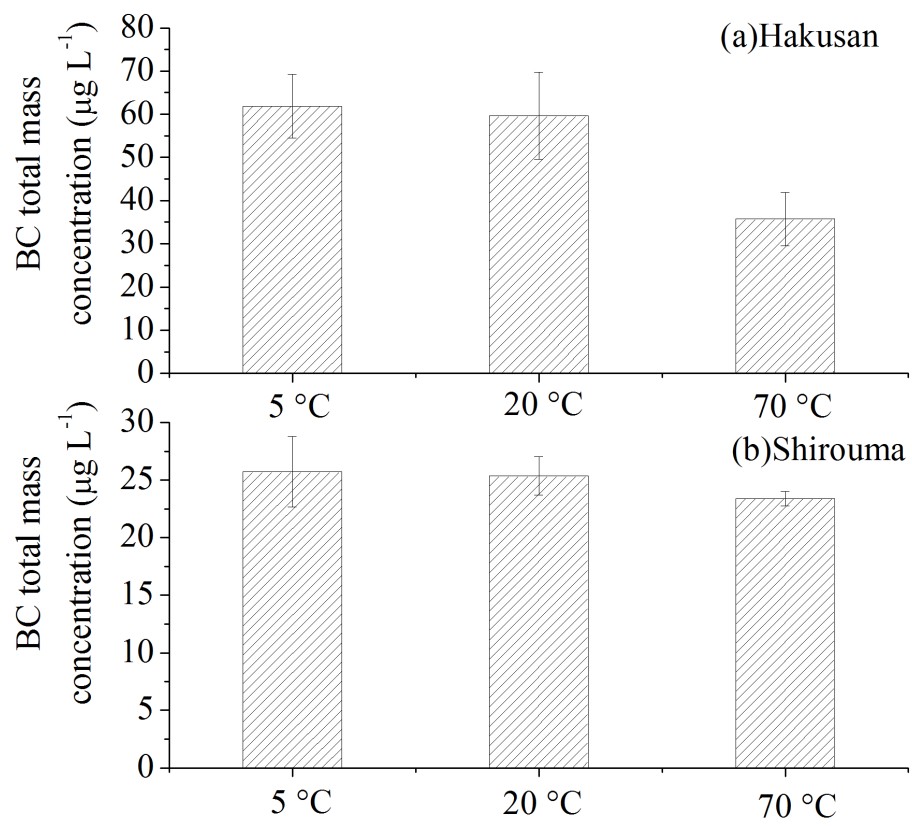

2    Figure 2. Total mass concentration of BC in (a) Hakusan and (b) Shirouma snow samples

3    melted at the temperatures of 5 ℃, 20 ℃ and 70 ℃.



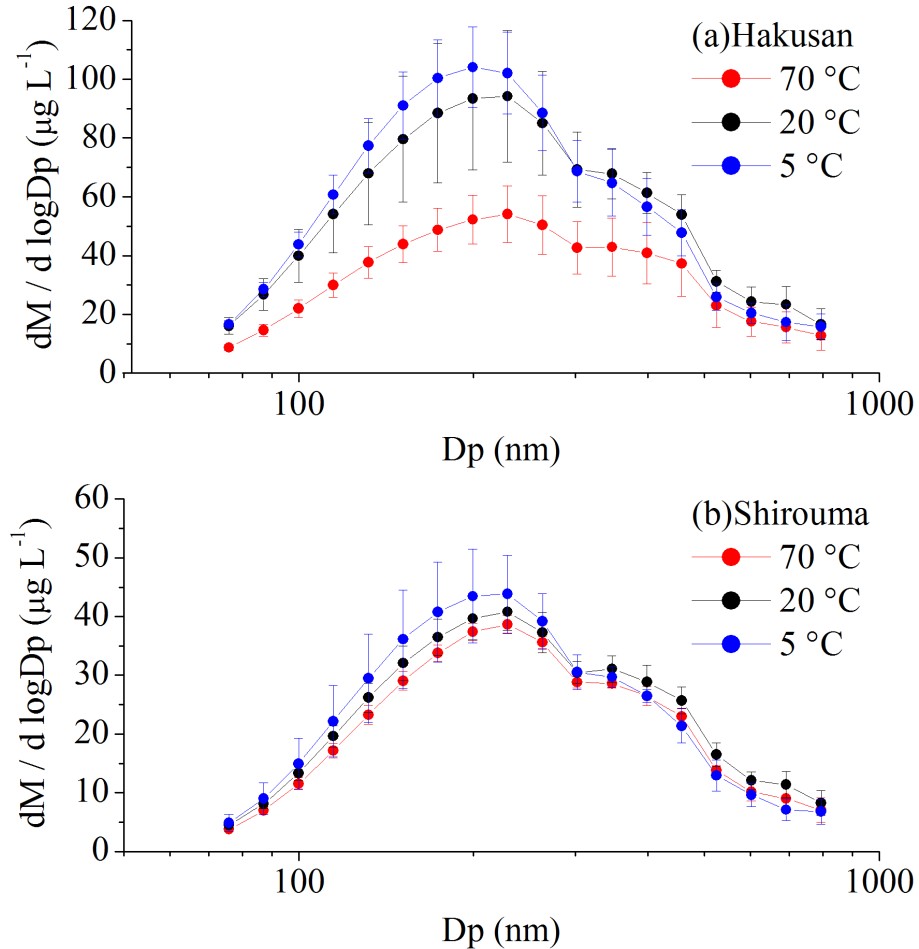

2      Figure .3 Mass size distribution of BC in (a) Hakusan and (b) Shirouma snow samples melted

3      at the temperatures of 5 ℃, 20 ℃ and 70 ℃, where M and Dp denote the mass and the

4      diameter of a BC particle, respectively.





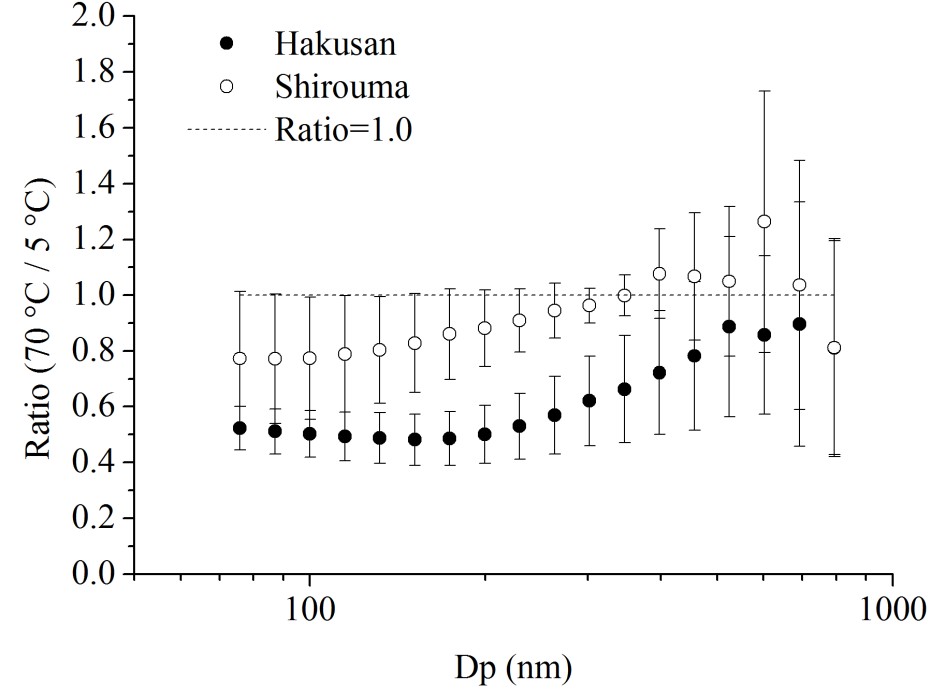

2     Figure 4. Size distribution of the ratio of the BC mass concentration in the snow samples

3     melted at 70 °C to that at 5 °C.




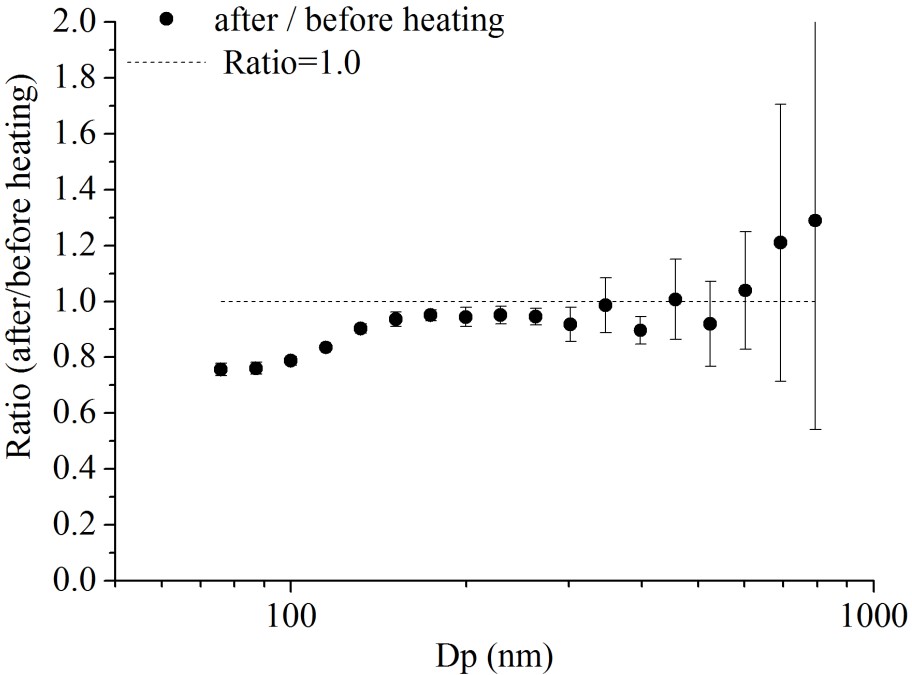

2    Figure 5. Size distribution of the ratio of the BC mass concentration in the Shirouma snow

3    sample melted at 5 °C and heated at 70 °C to that before the heating.





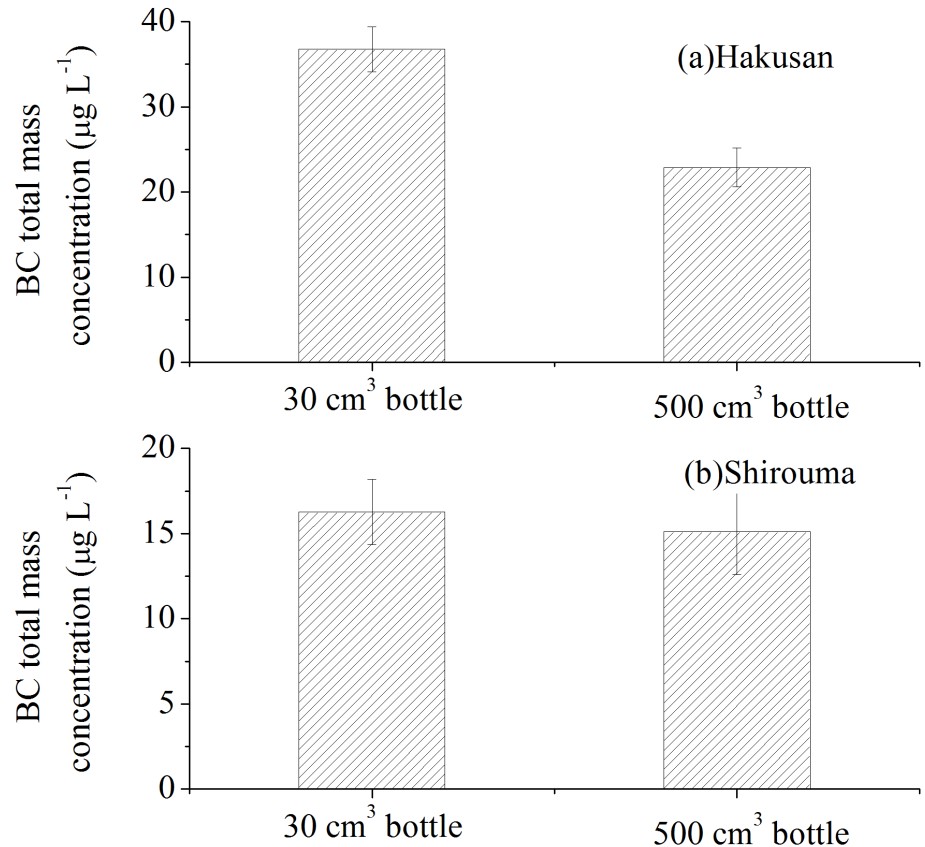

2    Figure 6. Total mass concentration of BC in (a) Hakusan and (b) Shirouma snow samples

3    melted in the 30 cm³ bottles and in the 500 cm³ bottle.





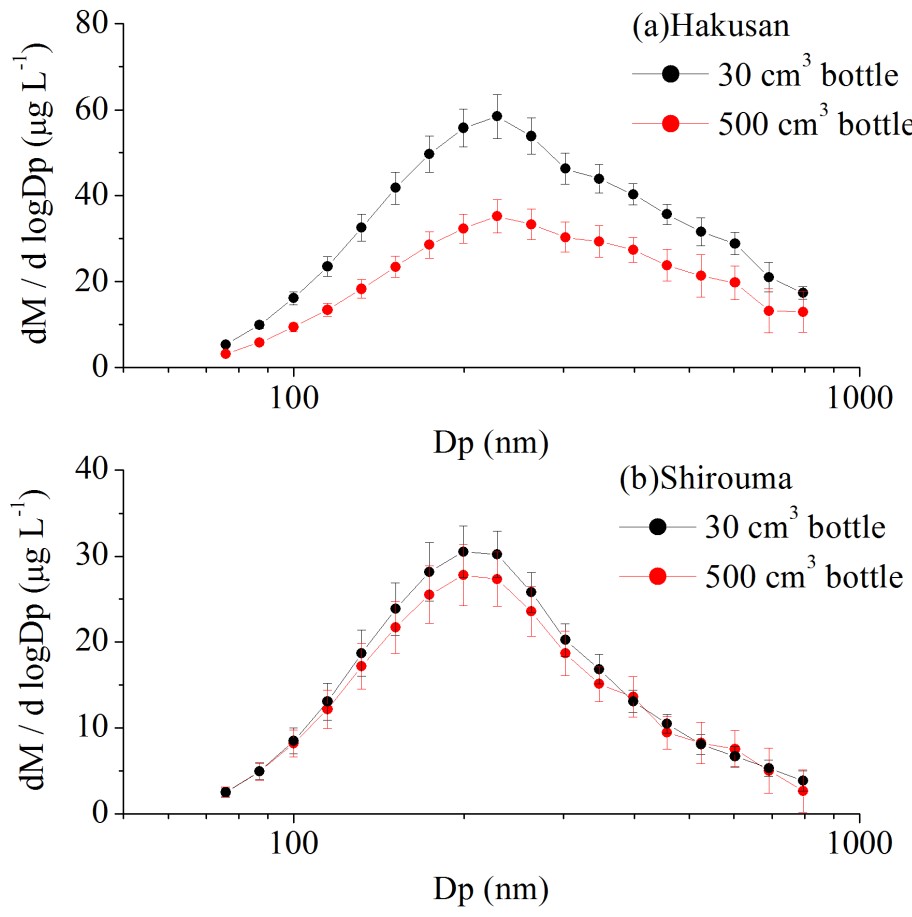

2   Figure 7. Size distribution of the BC mass concentration in (a) Hakusan and (b) Shirouma

3   snow samples melted in the 30 cm³ bottles and in the 500 cm³ bottle.





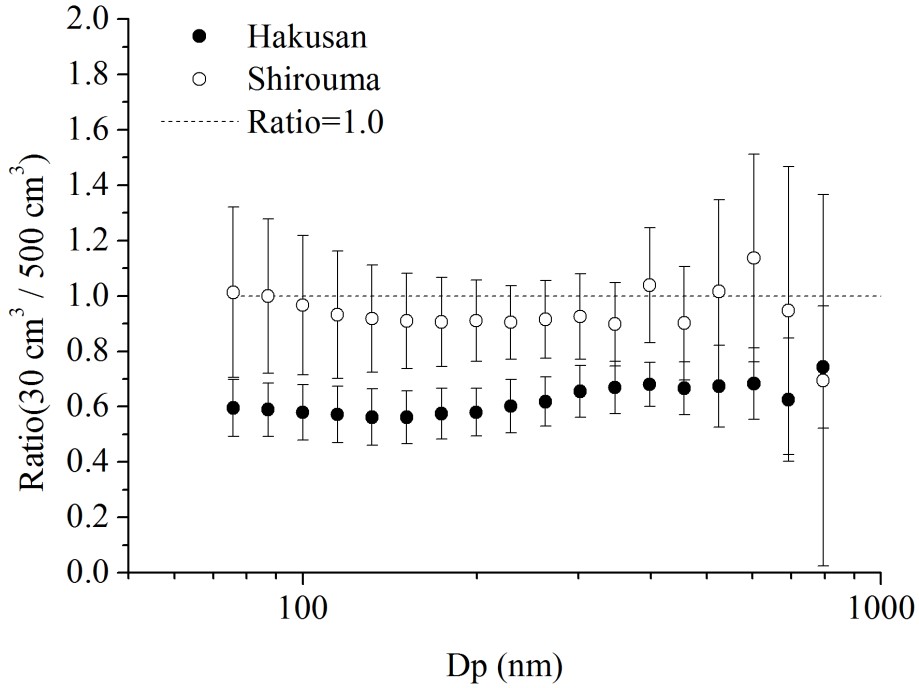

2    Figure 8. Size distribution of the ratio of the BC mass concentration in the snow sample

3    melted in the 500 cm$^3$ bottle to that melted in the 30 cm$^3$ bottles.

