# Peer review of "Influence of the melting temperature on the measurement"

_Atmospheric Measurement Techniques, 2015_

## Referee Comment (RC1) · Anonymous Referee #1 · 21 Jan 2016

General comments:

The paper reports results from a study of the effects of melt temperature and the amount of time over which a snow sample is melted on the mass and size distribution of BC in one set of new-snow samples and one set of aged-snow samples. The study is conducted using SP2 measurements of BC mass size distributions.

The scope of the study is somewhat limited. However, given the growing use of the SP2 to measure BC in snow samples, and the challenges of doing so – e.g. as noted by Schwarz et al. (2012) and Lim et al (2014), both referenced in the manuscript – I believe it should be published, and AMT is certainly the appropriate journal.
[Figure]

The paper is understandable as-written but would benefit from editing by a native English speaker. The Conclusion section in particular is in need of editing for clarity/flow.

Specific comments:

1.) Section 1, pg 3 paragraph starting on line 6 notes that previous studies have melted snow samples by heating in a microwave and by heating in a warm-water bath. The stated goal of the study is to test whether these approaches affect the size distribution and mass of BC in the melted snow sample. However, the study only tests for the effect of using different temperature water baths. There are no measurements of snow samples melted in a microwave oven. Tests would need to be done to see whether it matters, for example, if the snow is microwaved just long enough to melt the snow or long enough to actually warm the snow meltwater much above freezing. As no such studies are done, the authors should be clear that the results presented only apply to samples heated in a warm bath. This point should be made in the conclusions.

2.) pg. 4, lines 26-27: "Inhomogeneity in each snow sample was estimated with the standard deviation of measurement results for these three bottled samples melted at a same temperature." Then again on pg 7, lines 7-8, it is noted that three samples are used to determine error bars. An n of 3 is not sufficient to calculate a standard deviation. An alternative possibility: Instead of showing error bars in the figures using standard deviations for 3 samples that the relevant figures simply show all three values as, e.g., dots. Similarly, on pg 6, line 23 and in Figure 5, it is not clear if the error bars are again standard deviations of n=3 tests. If so, again, I think these should not be presented as standard deviations but instead show all three data points, as well as the mean. If it's not from n=3 tests, what is it?

3.) pg. 5, lines 15-17: snow melt-water samples were aerosolized with "a concentric pneumatic nebulizer (Marin-5, Cetac Technologies Inc., Omaha, Nebraska, USA), with a peristaltic pump (REGRO Analog, ISMATEC SA., Feldeggstrasse, Glattbrugg, Switzerland)". Schwarz et al. (2012) and Lim et al. (2014) have demonstrated variable

efficiencies for getting BC into the SP2 from liquid samples using different nebulizers. Was the efficiency of the system used here tested/quantified? If there is poor efficiency at larger sizes this could affect the conclusions about the change in total BC mass with heating temperature/rate. This is an important point that must be addressed, or at least acknowledged as a source of uncertainty in the study.

4.) Figure 1 & Table 1: I don't think Figure 1 is really needed. In the context of this study what is important is that the samples were of new and aged snow. No statement is made about how the geographic location of the samples might affect the study, so I would delete this map and just give the lat/lon of the sample locations for the interested reader. This is currently done in Table 1. The information contained in Table 1 is mostly also in the text. I think this information should be provided either in a table or in the text, but not both, given the brevity of this study/paper. My suggestion would be to delete both Figure 1 and Table 1, and simply include the relevant information in the text.

5.) It is not at all clear what the SO4, NO3 and other chemical analyses add to this study. They do not provide any information regarding whether or why the BC concentrations and size distributions are affected by the heating temperature or melt time. All reference to these analyses should be removed from the paper.

Technical corrections:

6.) Section 1: References to Bond et al. (2012) need to be corrected to Bond et al. (2013). This correction also needs to be made in the References list.

7.) pg. 2, lines 15-16: Bond et al. (2013) also provided a central estimate of 0.04 W/m2, not just a min/max.

8.) pg. 4-5, Section 2.2: Multiple references to "grass bottles" need to be corrected to "glass bottles"

9.) pg 6, lines 14-16: "Figure 4 shows the size distributions of the 30 BC mass ratio of the 70 °C melting sample to the 5 °C melting sample, indicating that the ratio systematically decreases with the decrease of the BC particle diameter." Suggest rewording to: "Figure 4 shows the ratio of BC mass in the samples heated to 70degC to those heated to 5degC, as a function of BC size. This shows that the ratio is lower for smaller particle sizes".

10.) pg. 6, lines 18-20: "...considering that the Hakusan sample was more aged and that it contained more pollutants such as SO42- and NO3- in comparison with the Shirouma sample." The Hakusan sample didn't only have higher SO4 an NO3 concentrations – it also has more than double the BC concentrations, as shown in Figure 3. Why not just state this directly?

11.) pg. 8, lines 25-27: "These results indicate that the decrease by the heating to high temperature can occur not only during the snow melting but also during the storage in the liquid phase." This statement needs to be modified: the decrease in mass was not for samples that were simply stored in liquid form, but that were heated to 70deg C (which is very warm, and so not a temperature samples would encounter simply by being stored at e.g. room temperature).

12.) pg. 8, lines 28-30: "In the melting time experiment, the Hakusan and Shirouma snow samples in the 30 cm3 bottles were melted for about 2 hours, and those in the 500 cm3 bottles were melted for more than 6 hours." This wording, and the discussion that follows, implies that this study was about the bottle size, not the amount of time it takes to melt smaller vs. larger snow samples. This sentence should be reworded, e.g. to: "The effect of melting time was also tested using the Hakusan and Shirouma snow samples. Sub-samples of each of approximately 30 cm3 took about 2 hours to melt at 1deg C, whereas samples of approximately 500 cm3 took about 6 hours to melt."

---

## Referee Comment (RC2) · Anonymous Referee #2 · 2 Feb 2016

Kinase et al have written an interesting paper on the influence of the melting temperature on black carbon measurements when processing snow samples with the SP2 instrument. Given that this technique is being used more frequently, I feel that the article is suitable for publication in AMT. Although I feel that it is suitable for publication, there are several issues that I feel need to be addressed, or at least mentioned.

1. The study size is very small. Snow samples collected from two locations only were analyzed. And, three replicates of each procedure were analyzed. The results for the two samples locations varied significantly. This would suggest that a more systematic study is clearly needed to quantify the loss as a function of melting temperature as well as snow conditions. The authors should state this.
[Figure]

2. The study is somewhat limited in that the samples are melted in "specific temperature" water baths. If the sample is large enough, it is possible that the actual temperature of the sample does not rise to the temperature of the bath. This is relevant to techniques which use larger sample volumes and filtering techniques (eg the University of Washington technique as well as the technique described in Schmitt et al: http://www.the-cryosphere.net/9/331/2015/tc-9-331-2015.html).

3. The article could benefit from editing by an English language expert. There are numerous statements that are either grammatically incorrect or awkward and it is necessary to clear up those issues before publication.

4. Did the authors quantify dust at all? The aged sample is likely to contain a lot more dust than the fresh snow sample.

Minor items: Page 1 line 18: Change "time conditions" to "amounts of time". Line 18: remove "its". Then line 19 change "distribution" to "distributions" Line 20: change to "The experiments where the melting temperatures were varied. ." Page 2 line 1: change "or in a long time" to "or over a long time period" Note: After the abstract, I won't address each grammar error individually. Line 11: it should also be noted in the publication that albedo changes can lead to significant changes in timing of snow melt therefore affecting water supply, therefore BC on snow isn't solely a climate issue. Line 21: light "transmission", not "transparency" Line 23: the second "technique" (not "one") Page 3, line 5: There have been a few intercomparison studies between techniques (Schwarz et al, 2012). It might be of value to mention these studies and a brief summary of their uncertainties in order to further support the need for understanding all aspects of the techniques. Line 6-31: The authors comment several times about the melting of snow samples using a microwave oven. Clearly the process of melting is somewhat different using a microwave oven versus using a warm water bath. The study only involves using a warm water bath for melting. This should be stated. Page 4, line 10: instead of "it was", "the snow samples were" Page 5, lines 29-30: Can you speculate as to why the uncertainties in the Hakusan samples were so much larger
than in the Shirouma samples? Page 6 line 17: consider changing "presumably depends" to "could depend". Line 25: Looking at the graph in Figure 5, it seems that the substantial loss begins around 150 nm rather than 300. Page 7 line 30: use "after" rather than "since" Page 8 line 1: On successive days, were the samples stirred or shaken? (This is stated in the conclusions, but should be stated earlier)

---

## Short Comment (SC1) · 2 Feb 2016

This paper raises an important issue regarding the reliability and quality of BC measurements in snow (and ice). While the paper certainly presents interesting results some details of methodological information are missing although they are substantial to allow interpretation of the presented results. The following lists the details of information which remain unclear in the current manuscript and discusses why they are relevant for the conclusions made.

(1) What standard material has been used for calibration of the results? The choice of standard material has been discussed in depth previously (Wendl et al., AMT, 2014). This study definitely should be cited in the manuscript since standardization and calibration is an important issue when analyzing BC in snow and ice. I am aware that the description of standard used in the study discussed here can be found in the literature cited (method section) but I do not think this is sufficient. Other related and important questions are: How was the calibration performed? Once a day, weekly...? Most importantly, do the discussed effects also show up in standard samples (e.g. did you analyze standard solutions which were treated similar to the individual sets discussed for comparison/as a reference)? How can the observed differences/similarities between standard and sample behavior be explained?

(2) It is unclear if the containers containing the snow samples were kept closed during melting. I assume they were, but still it should be described clearly in the manuscript. This is important since if they were open, effects due to evaporation cannot be excluded. Evaporation would likely result in an increase in concentration but might not be detected because being masked by other, larger effects having an opposite direction (i.e. resulting in decreased concentration) which then might be underestimated as a consequence.

(3) It remains unclear if once the snow samples were melted they were sonicated in the containers used for melting or if aliquots were first taken, transferred to new containers and sonicated afterwards. If the latter is the case, any wall effects happening in the containers used to melt the samples will not be considered even though they are likely significant (decrease in the observed concentrations, likely size dependent) and would allow a different interpretation of the derived results. This should be clarified and also be discussed if necessary based on the procedure used. It should also be described if the samples were stirred prior/during sonification and/or analysis. This is of particular importance to know for the sample in the 500 cm3 bottle where sedimentation might have a significant effect resulting in the difference observed compared to the 30 cm3 containers.

(4) Further, what was the time passing between sonification and analysis of samples? Was it similar for all samples? Again this information is relevant because wall effects happening between sonification and analysis can result in the observation of decreased concentrations. If there is a large deviation in the time passed between time of sonification and analysis for the individual sample such wall effects might contribute significantly to any observed decrease in concentrations. Has this been tested? This should be clarified and discussed.

I am fully aware that the author's might have omitted this degree of detailed information because some of it seems rather trivial. Nevertheless, I strongly believe it needs to be addressed carefully for the reasons pointed out.

More general and not considering the above, the fact that only two different samples were used for the investigations seems limiting to reach sound conclusions. Are the samples studied here representative? Their concentration is not so different after all. The question remains if the investigated effects are also significant for samples with a much different BC particle size distribution or much lower/higher concentrations? In fact, even for the results presented here, the significance of the described effects may be questioned regarding the uncertainties (see e.g. Fig. 2) and the fact that the number of samples in each set is rather low (n = 3 for 30 cm3 containers and n = 1 for 500 cm3). Is this also the reason why there is not given a clear recommendation what temperature and melting time should preferentially be used in order to obtain the most reliable results?

Other remarks:

p.7, line 28 – p.8, line 6: Were the stored samples sonicated once again prior to each subsequent analysis? If this is not the case and still no effect of storage time was observed these results would be very different from what has been described in Wendl et al. (2014). Please comment/discuss.

p.2, line 23 ff.: If soot is measured with a thermal optical technique, it is referred to as EC (elemental carbon), not BC. The authors should also cite some of the pioneer work of such measurements in snow and ice:

Lavanchy, V.M.H., Gäggeler, H.W., Schotterer, U., Schwikowski, M. and Baltensperger, U. (1999). Historical record of carbonaceous particle concentrations from a European high-alpine glacier (Colle Gnifetti, Switzerland). Journal of Geophysical Research 104: doi: 10.1029/1999JD900408. issn: 0148-0227.

Jenk, T. M., Szidat, S., Schwikowski, M., Gaeggeler, H. W., Bruetsch, S., Wacker, L., Synal, H. A., and Saurer, M.: Radiocarbon analysis in an Alpine ice core: record of anthropogenic and biogenic contributions to carbonaceous aerosols in the past (1650-1940), Atmospheric Chemistry and Physics, 6, 5381-5390, 2006.

p.2, line 28 ff.: To be complete and because it discusses a lot of similar and complementary issues to the ones addressed in this paper, Wendl et al. which was actually published in the same Journal as this study(!) needs to be cited and should also be discussed (see above):

Wendl, I. A., Menking, J. A., Färber, R., Gysel, M., Kaspari, S. D., Laborde, M. J. G., and Schwikowski, M.: Optimized Method for Black Carbon Analysis in Ice and Snow Using the Single Particle Soot Photometer, Atmos. Meas. Tech., 7, 2667–2681, 2014, doi:10.5194/amt-7-2667-2014.

---

## Author Comment (AC1) · 13 Mar 2016

The authors thank the reviewer for his/her comments helpful and useful for improving our manuscript. Replies to each comment are shown as the followings:

1.) Section 1, pg 3 paragraph starting on line 6 notes that previous studies have melted snow samples by heating in a microwave and by heating in a warm-water bath. The stated goal of the study is to test whether these approaches affect the size distribution and mass of BC in the melted snow sample. However, the study only tests for the effect of using different temperature water baths. There are no measurements of snow samples melted in a microwave oven. Tests would need to be done to see whether it matters, for example, if the snow is microwaved just long enough to melt the snow or

long enough to actually warm the snow melt water much above freezing. As no such studies are done, the authors should be clear that the results presented only apply to samples heated in a warm bath. This point should be made in the conclusions.

(Ans.1)In this study, we used water bath in order to specify the melting temperature because it is difficult to control and specify the temperature when we use a microwave oven. Prior to the experiments shown in this manuscript, as a trial, we measured and compared the BC mass concentrations in two parts of the Shirouma and Hakusan samples: ones had been melted at the room temperature and the others had been melted with a microwave oven. The BC mass concentrations in the samples melted with the microwave oven were significantly reduced. Thus we think the melting using a microwave oven heat the snow sample to high temperature enough to influence BC measurement. To show them, we have modified our manuscript.

2.) pg. 4, lines 26-27: "Inhomogeneity in each snow sample was estimated with the standard deviation of measurement results for these three bottled samples melted at a same temperature." Then again on pg 7, lines 7-8, it is noted that three samples are used to determine error bars. An n of 3 is not sufficient to calculate a standard deviation. An alternative possibility: Instead of showing error bars in the figures using standard deviations for 3 samples that the relevant figures simply show all three values as, e.g., dots. Similarly, on pg 6, line 23 and in Figure 5, it is not clear if the error bars are again standard deviations of n=3 tests. If so, again, I think these should not be presented as standard deviations but instead show all three data points, as well as the mean. If it's not from n=3 tests, what is it?

(Ans.2)We agree the reviewer's comment, and have modified the text and figures to adopt median with range between minimum and maximum values to the representative value and variability instead of the average and standard deviation.

3.) pg. 5, lines 15-17: snow melt-water samples were aerosolized with "a concentric pneumatic nebulizer (Marin-5, Cetac Technologies Inc., Omaha, Nebraska, USA),

with a peristatic pump (REGRO Analog, ISMATEC SA., Feldeggstrasse, Glattbrugg, Switzerland)". Schwarz et al. (2012) and Lim et al. (2014) have demonstrated variable efficiencies for getting BC into the SP2 from liquid samples using different nebulizers. Was the efficiency of the system used here tested/quantified? If there is poor efficiency at larger sizes this could affect the conclusions about the change in total BC mass with heating temperature/rate. This is an important point that must be addressed, or at least acknowledged as a source of uncertainty in the study.

(Ans.3)As pointed out by the reviewer, it is possible that a poor efficiency may affect the result. Our experimental system was nearly identical with that in Mori et al. (2106), who estimated the extraction efficiency of a Marin-5 nebulizer to be mostly 50% in the diameter range of 200–2000 nm, and that the efficiency was stable. This value is much higher than other nebulizers, such as a U-5000AT and a collision-type nebulizer (Ohata et al., 2012, 2013; Schwartz et al., 2012). Although a APEX-Q nebulizer has higher efficiencies up to 72 % in the diameter range of 150–600 nm (Lim et al., 2014) the efficiency of this nebulizer depends on the diameter between 100 and 1000 nm (Wendl et al., 2014). Therefore, we adopted the Marin-5 nebulizer in this study, and we did intercomparison between our system and that in Mori et al. to show the efficiency of two systems agreed within their random error range. We have modified our manuscript to show these.

4.) Figure 1 & Table 1: I don't think Figure 1 is really needed. In the context of this study what is important is that the samples were of new and aged snow. No statement is made about how the geographic location of the samples might affect the study, so I would delete this map and just give the lat/lon of the sample locations for the interested reader. This is currently done in Table 1. The information contained in Table 1 is mostly also in the text. I think this information should be provided either in a table or in the text, but not both, given the brevity of this study/paper. My suggestion would be to delete both Figure 1 and Table 1, and simply include the relevant information in the text.

(Ans.4)Following the reviewer's comment, and have modified the manuscript.

[Figure]

5.) It is not at all clear what the SO4, NO3 and other chemical analyses add to this study. They do not provide any information regarding whether or why the BC concentrations and size distributions are affected by the heating temperature or melt time. All reference to these analyses should be removed from the paper.

(Ans.5)As pointed out in this reviewer's comment, the ion measurement results were not used in the interpretation of the experimental results. We have modified the manuscript to remove information on ion analysis from the snow sample section. However, some studies showed possibilities that ion components may influence the BC measurement and BC size distribution, and the ion measurement showed that amounts of some ions were clearly different in the two samples. We also modified the manuscript to show the possibility that the impurities in snow may be significant in the conclusion section.

Technical corrections: Thanking these reviewer's technical comments, we have modified the manuscript.

6.) Section 1: References to Bond et al. (2012) need to be corrected to Bond et al. (2013). This correction also needs to be made in the References list. 7.) pg. 2, lines 15-16: Bond et al. (2013) also provided a central estimate of 0.04 W/m2, not just a min/max. 8.) pg. 4-5, Section 2.2: Multiple references to "grass bottles" need to be corrected to "glass bottles" 9.) pg 6, lines 14-16: "Figure 4 shows the size distributions of the 30 BC mass ratio of the 70 _C melting sample to the 5 _C melting sample, indicating that the ratio systematically decreases with the decrease of the BC particle diameter." Suggest rewording to: "Figure 4 shows the ratio of BC mass in the samples heated to 70degC to those heated to 5degC, as a function of BC size. This shows that the ratio is lower for smaller particle sizes". 12.) pg. 8, lines 28-30: "In the melting time experiment, the Hakusan and Shirouma snow samples in the 30 cm3 bottles were melted for about 2 hours, and those in the 500 cm3 bottles were melted for more than 6 hours." This wording, and the discussion that follows, implies that this study was about the bottle size, not the amount of time it takes to melt smaller vs. larger snow

samples. This sentence should be reworded, e.g. to: "The effect of melting time was also tested using the Hakusan and Shirouma snow samples. Sub-samples of each of approximately 30 cm3 took about 2 hours to melt at 1deg C, whereas samples of approximately 500 cm3 took about 6 hours to melt." 10.) pg. 6, lines 18-20: ": : :considering that the Hakusan sample was more aged and that it contained more pollutants such as SO42- and NO3- in comparison with the Shirouma sample." The Hakusan sample didn't only have higher SO4 an NO3 concentrations – it also has more than double the BC concentrations, as shown in Figure 3. Why not just state this directly?

(Ans.6)As pointed out in this reviewer's comment, higher BC concentration indicated the Hakusan snow sample was more polluted than the Shirouma sample. We have modified the manuscript to show it, and that the difference in the BC measurement influences between the two samples could not explain only by the BC amount itself.

11.) pg. 8, lines 25-27: "These results indicate that the decrease by the heating to high temperature can occur not only during the snow melting but also during the storage in the liquid phase." This statement needs to be modified: the decrease in mass was not for samples that were simply stored in liquid form, but that were heated to 70deg C (which is very warm, and so not a temperature samples would encounter simply by being stored at e.g. room temperature).

(Ans.7)The experiment that the heating the liquid to 70C indicate not only the influence of high temperature which the liquid would not encounter by storage at room temperature, but also that the BC decrease and size distribution modification under a high temperature could occur in the liquid phase. We have modified the manuscript for indicating them more clearly.

(a)Hakusan

(b)Shirouma

**Fig. 1.**

[Figure]

**Fig. 2.**

[Figure]

[Figure]

**Fig. 3.**

[Figure]

**Fig. 4.**

**Fig. 5.**

(a)Hakusan

(b)Shirouma

*Y-axis (both plots):* BC total mass concentration ($\mu$g L$^{-1}$)

*X-axis (both plots):* 30 cm$^3$ bottle, 500 cm$^3$ bottle

Fig. 6.

(a)Hakusan

- 30 cm$^3$ bottle
- 500 cm$^3$ bottle

(b)Shirouma

- 30 cm$^3$ bottle
- 500 cm$^3$ bottle

dM / d logDp (µg L$^{-1}$)

Dp (nm)

[Figure]

Fig. 7.

---

## Author Comment (AC2) · 13 Mar 2016

The authors thank the detailed comments useful for improving our manuscript. Replies to each comment are shown as the followings:

(1) What standard material has been used for calibration of the results? The choice of standard material has been discussed in depth previously (Wendl et al., AMT, 2014). This study definitely should be cited in the manuscript since standardization and calibration is an important issue when analyzing BC in snow and ice. I am aware that the description of standard used in the study discussed here can be found in the literature cited (method section) but I do not think this is sufficient. Other related and important questions are: How was the calibration performed? Once a day, weekly: :

:? Most importantly, do the discussed effects also show up in standard samples (e.g. did you analyze standard solutions which were treated similar to the individual sets discussed for comparison/as a reference)? How can the observed differences/similarities between standard and sample behavior be explained?

(Ans.1)The experiment system used in this study was nearly identical with that in Mori et al. (2016) who conducted experiments to measure system efficiencies to evaluate absolute sensitivity using model BC particles. We had done the intercomparison of our system with that in Mori et al, (2016). before our study, and found the efficiency and sensitivity of the two systems agreed within their random error range, indicating that sensitivity of our system was also nearly identical with that in Mori et al. (2016). It should be noted that results in this study depends on only relative values, not on the absolute values. Therefore, stability of the system sensitivity is much more significant in this study, rather than absolute calibration. A temporal drift of the system sensitivity during the experiments had been examined by measuring some samples twice (time interval was 1 to 24 hours) in each experiment. The difference of the first and second measurement results was 2.1(0.4 to 5.1)% for 30 cm3 bottle samples, and it was 7.8% for a 500 cm3 sample, indicating that the temporal drift of the system sensitivity was less than or comparable with random errors. To show this result, we have modified the manuscript.

(2) It is unclear if the containers containing the snow samples were kept closed during melting. I assume they were, but still it should be described clearly in the manuscript. This is important since if they were open, effects due to evaporation cannot be excluded. Evaporation would likely result in an increase in concentration but might not be detected because being masked by other, larger effects having an opposite direction (i.e. resulting in decreased concentration) which then might be underestimated as a consequence.

(Ans.2)In order to avoid the contamination and evaporation effect as pointed out with this comment, we sealed containers with bottle cap during the melting, the storage and

the sonication. To show this, we have modified the manuscript.

(3) It remains unclear if once the snow samples were melted they were sonicated in the containers used for melting or if aliquots were first taken, transferred to new containers and sonicated afterwards. If the latter is the case, any wall effects happening in the containers used to melt the samples will not be considered even though they are likely significant (decrease in the observed concentrations, likely size dependent) and would allow a different interpretation of the derived results. This should be clarified and also be discussed if necessary based on the procedure used. It should also be described if the samples were stirred prior/during sonication and/or analysis. This is of particular importance to know for the sample in the 500 cm3 bottle where sedimentation might have a significant effect resulting in the difference observed compared to the 30 cm3 containers.

(Ans.3)After aliquots of the snow samples were taken in a bottle, the melting, the storage, the sonication and the BC analysis of the sample were conducted in the same bottle without a sample transfer to avoid loss or contamination. All sample bottles including 500 cm3 bottles were stirred by shaking them following the sonication just before the analysis. To show these clearly, we have modified the manuscript. In addition, we analyzed 500cm3 samples several time during the 4 days storage (shown in section 3.2), and no significant decrease was found during the storage. These showed that sedimentation effect was insignificant for our experiments.

(4) Further, what was the time passing between sonication and analysis of samples? Was it similar for all samples? Again this information is relevant because wall effects happening between sonication and analysis can result in the observation of decreased concentrations. If there is a large deviation in the time passed between time of sonication and analysis for the individual sample such wall effects might contribute significantly to any observed decrease in concentrations. Has this been tested? This should be clarified and discussed.

(Ans.4)The time passing between the sonication followed by the bottle shake and the analysis of the sample fluctuated between 1 and 10 minutes. We have not examined this time passing effect. However, as already written, some samples measured twice showed the difference of the two measurements, where the above time passing were different, was less than or comparable with random errors. Thus the influence of the time passing less than 10 minutes was less than or included in the random error.

I am fully aware that the author's might have omitted this degree of detailed information because some of it seems rather trivial. Nevertheless, I strongly believe it needs to be addressed carefully for the reasons pointed out. More general and not considering the above, the fact that only two different samples were used for the investigations seems limiting to reach sound conclusions. Are the samples studied here representative? Their concentration is not so different after all. The question remains if the investigated effects are also significant for samples with a much different BC particle size distribution or much lower/higher concentrations? In fact, even for the results presented here, the significance of the described effects may be questioned regarding the uncertainties (see e.g. Fig. 2) and the fact that the number of samples in each set is rather low (n = 3 for 30 cm3 containers and n = 1 for 500 cm3). Is this also the reason why there is not given a clear recommendation what temperature and melting time should preferentially be used in order to obtain the most reliable results?

(Ans.5)We think that it is not very significant that the samples used in this study were representative. (It is not reasonable to think that only two samples represent all snow samples.) Our experiments, which detected the influence of the melting temperature and time adopting only the two samples, strongly suggest that similar influence could occur in more various snow samples. The difference in the two samples indicated that these influences depends on the snow conditions, and this dependence makes difficult to determine clear, preferable conditions for the melting snow sample, as pointed out by this comment. The study enable us notice of significance of the lower melting temperature and shorter melting time for accurate measurement of BC in snow, and

recommends us to test that the adopted melting temperature and time do not affect the measurement for the snow sample. To show this, we have modified the manuscript.

Other remarks: p.7, line 28 – p.8, line 6: Were the stored samples sonicated once again prior to each subsequent analysis? If this is not the case and still no effect of storage time was observed these results would be very different from what has been described in Wendl et al. (2014). Please comment/discuss.

p.2, line 23 ff.: If soot is measured with a thermal optical technique, it is referred to as EC (elemental carbon), not BC. The authors should also cite some of the pioneer work of such measurements in snow and ice: Lavanchy, V.M.H., Gäggeler, H.W., Schotterer, U., Schwikowski, M. and Baltensperger, U. (1999). Historical record of carbonaceous particle concentrations from a European high-alpine glacier (Colle Gnifetti, Switzerland). Journal of Geophysical Research 104: doi: 10.1029/1999JD900408. issn: 0148-0227. Jenk, T. M., Szidat, S., Schwikowski, M., Gaeggeler, H. W., Bruetsch, S., Wacker, L., Synal, H. A., and Saurer, M.: Radiocarbon analysis in an Alpine ice core: record of anthropogenic and biogenic contributions to carbonaceous aerosols in the past (1650- 1940), Atmospheric Chemistry and Physics, 6, 5381-5390, 2006.

p.2, line 28 ff.: To be complete and because it discusses a lot of similar and complementary issues to the ones addressed in this paper, Wendl et al. which was actually published in the same Journal as this study(!) needs to be cited and should also be discussed (see above): Wendl, I. A., Menking, J. A., Färber, R., Gysel, M., Kaspari, S. D., Laborde, M. J. G., and Schwikowski, M.: Optimized Method for Black Carbon Analysis in Ice and Snow Using the Single Particle Soot Photometer, Atmos. Meas. Tech., 7, 2667–2681, 2014, doi:10.5194/amt-7-2667-2014.

Thanking these comments, we have modified the manuscript to quote these references.

---

## Author Comment (AC3) · 13 Mar 2016

The authors thank the reviewer for his/her comments helpful and useful for improving our manuscript. Replies to each comment are shown as the followings:

1.) The study size is very small. Snow samples collected from two locations only were analyzed. And, three replicates of each procedure were analyzed. The results for the two samples locations varied significantly. This would suggest that a more systematic study is clearly needed to quantify the loss as a function of melting temperature as well as snow conditions. The authors should state this.

(Ans.1)Following this comment, we have modified the manuscript to include this state-

ment in the conclusion section.

2.) The study is somewhat limited in that the samples are melted in "specific temperature" water baths. If the sample is large enough, it is possible that the actual temperature of the sample does not rise to the temperature of the bath. This is relevant to techniques which use larger sample volumes and filtering techniques (eg the University of Washington technique as well as the technique described in Schmitt et al: http://www.the-cryosphere.net/9/331/2015/tc-9-331-2015.html).

(Ans.2)It would take very long time to melt a large volume of snow sample, as used in the filtering techniques, especially at a low temperature as recommended in this study. This long melting time could affect the BC measurement result as shown in the melting time experiment. This is one of the reasons why the SP2 technique is more adequate for measuring BC in snow. We have modified the manuscript to show this. If a large volume sample would be melt at a high temperature, a considerable part of the sample would be heated near the container wall, the BC decrease could partly occur.

3.) The article could benefit from editing by an English language expert. There are numerous statements that are either grammatically incorrect or awkward and it is necessary to clear up those issues before publication.

(Ans.3)Following this comment, the revised manuscript has been edited by an English language expert.

4.) Did the authors quantify dust at all? The aged sample is likely to contain a lot more dust than the fresh snow sample.

(Ans.4)We did not measured dust in the snow samples in this study. This comment suggests us that dusts in snow may cause the BC decrease by adsorption on their surface. This is only speculation, and a systematic study is necessary to understand a role of dusts. We have modified the manuscript to show this.

Minor items: Thanking these reviewer's comments, we have modified the manuscript.

Page 1 line 18: Change "time conditions" to "amounts of time". Line18: remove "its". Then line 19 change "distribution" to "distributions" Line 20: change to "The experiments where the melting temperatures were varied. ." Page 2 line 1: change "or in a long time" to "or over a long time period" Note: After the abstract, I won't address each grammar error individually. Line 11: it should also be noted in the publication that albedo changes can lead to significant changes in timing of snow melt therefore affecting water supply, therefore BC on snow isn't solely a climate issue. Line 21: light "transmission", not "transparency" Line 23: the second "technique" (not"one") Page 3, line 5: There have been a few intercomparison studies between techniques (Schwarz et al, 2012). It might be of value to mention these studies and a brief summary of their uncertainties in order to further support the need for understanding all aspects of the techniques. Page4, line 10: instead of "it was", "the snow samples were" Page 6 line 17: consider changing "presumably depends"to "could depend". Page 7 line 30: use "after" rather than "since" Page 8 line 1: On successive days, were the samples stirred or shaken? (This is stated in the conclusions, but should be stated earlier) Page 3, Line 6-31: The authors comment several times about the melting of snow samples using a microwave oven. Clearly the process of melting is somewhat different using a microwave oven versus using a warm water bath. The study only involves using a warm water bath for melting. This should be stated.

(Ans.5)In this study, we used water bath in order to specify the melting temperature because it is difficult to control and specify the temperature when we use a microwave oven. Because melting using a microwave oven heats the snow sample at temperatures significantly higher than room temperature, it could influence the BC measurement. We have modified the manuscript to show them.

Page 5, lines 29-30: Can you speculate as to why the uncertainties in the Hakusan samples were so much larger than in the Shirouma samples?

(Ans.6)The higher uncertainties in the Hakusan sample would be mainly attributed to a larger inhomogeneity in the sample. The Hakusan sample was aged, and the
snow grain size was much larger than that of the Shirouma sample, indicating that the Hakusan snow had experienced the partial melting and re-freezing of snow. Re-distribution of impurities in snow probably occurred during these processes to increase their inhomogeneity.

Line 25: Looking at the graph in Figure 5, it seems that the substantial loss begins around 150 nm rather than 300.

(Ans.7)As pointed out by the reviewer, the substantial loss exceeding 10% begins around 186 nm. However, the loss found between 186 and 324 nm is also signifi-cant, exceeding a random error range as shown in Figure 5 (Figure 4 in the revised manuscript). We have modified the manuscript to show them more clearly.
* * *